# Spiritual Legitimacy in Contemporary Japan: A Case Study of the Power Spot Phenomenon and the Haruna Shrine, Gunma

Shin Yasuda 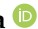

Faculty of Regional Policy, Takasaki City University of Economics, Gunma 3700801, Japan; syasuda@tcue.ac.jp

**Abstract:** Since the 2000s, Japanese internet media as well as mass media, including magazines, television and newspapers, have promoted the concept of a "power spot" as part of the spirituality movement in the country. This emerging social environment for the power spot phenomenon has developed a new form of religiosity, which can be called "spiritual legitimacy," according to the transformation of religious legitimacy embedded in Japanese society. This paper, therefore, examined the emergence of a new form of spiritual legitimacy utilizing a case study of the power spot phenomenon in the Haruna Shrine, Gunma Prefecture, in Japan. The development of the power spot phenomenon in the Haruna Shrine indicates that consumption of spiritual narratives has strongly promoted the construction of a social context of spiritual legitimacy, such as through shared images and symbols related to the narratives in the sacred site. As a result, this paper clarifies that this new form of spiritual legitimacy embodies stakeholders' social consensus on spiritual narratives, which people have struggled to construct a social context for spiritual legitimacy to ensure hot authentication of their individual narratives and experiences.

**Keywords:** power spot; spirituality; social context; spiritual legitimacy; Japan



## 1. Introduction

Since the 2000s, Japanese internet media, such as webpages, blogs, social networking services (SNS) and mass media, such as magazines, television and newspapers, have promoted the concept of a "power spot" or a "spiritual spot"—a place believed to enhance one's spiritual fulfilment and wellness, and offers healing, health, and good fortune (Horie 2009, 2017; Suga 2010; Uchikawa 2017; Kato and Pregano 2017; Carter 2018). Although the term "power spot" has been used since the 1980s by New Age movements worldwide (Horie 2009), and other related movements in Japan (Horie 2017), its popularity has increased among contemporary Japanese through mass media and tourism industries (Kodera 2011; Carter 2018; Nakanishi 2018; Okamoto 2020; Suzuki 2020). Carter (2018) explained that the contemporary power spot phenomenon is strongly connected with tourism and mass media as well as popular discourses on spirituality, nature and sacred sites (Carter 2018, p. 148). Visits to power spot destinations have become one of the major motivations for the Japanese, including those without a particular religious denomination or preferences (Okamoto 2015, 2020; Yamaguchi 2017; Carter 2018; Suzuki 2020; Tillonen 2021). With the development of the power spot phenomenon in Japan, traditional religious sites such as Shinto shrines and Buddhist temples have been transformed to suit the narrative of the power spot (Horie 2017; Carter 2018; Nakanishi 2018; Tillonen 2021).

In general, contemporary Japanese society is reluctant to embrace religion in daily life and identifies as secular (Reader 2012). However, the concept of power spot has been widely accepted as an alternative to traditional religious practices and institutions (Horie 2009, 2017; Okamoto 2015, 2020). Local administrations began to actively promote religious sites as destinations for power spot experiences as well as for their cultural and historical significance (Uchikawa 2017; Suzuki 2020). Moreover, local residents and visitors began to demonstrate their individual piety and commitment towards these sacred places in the

form of a power spot narrative rather than as religious devotion (Kato and Pregano 2017; Pregano and Kato 2020). These stakeholders began to organize local events that promoted power spot experiences and attracted a wide range of visitors (Okamoto 2015, pp. 123–81).

Although the power spot phenomenon is widely accepted in contemporary Japanese society, the concept of a power spot is not rigidly defined (Kodera 2011; Horie 2017; Uchikawa 2017; Carter 2018). The term is loosely referred to as travel "to a place thought to embody spiritual energies from the earth" (Carter 2018, p. 148), and power spots are variably referred to as "the place where energy which influences individual body and mind emerges" (Kodera 2011, p. 87); "the place where one feels power with healing, health, and fortune" (Uchikawa 2017, p. 61); or "specific places where energy gathers and bestows good fortune, healing or other practical benefits on visitors" (Tsukada and Ōmi 2011; Kato and Pregano 2017, p. 246). Some narratives explain the characteristics of a power spot from the perspective of *qigong/kikō* (Chinese and Japanese life-energy cultivation), *feng shui/fūsui* (Chinese and Japanese geomancy), and other Japanese folk beliefs, while other narratives do not have a certain origin (Carter 2018, p. 153). Most narratives do not offer a clear definition of the concept; rather, they are based on individual preferences, attitudes and understandings of a certain religiosity or spirituality. Kato and Pregano (2017) wrote that the characteristics of the power spot phenomenon in Japan can be considered to have wider connotations beyond religion, including health, wellness, and self-improvement (Kato and Pregano 2017, p. 243).

In this environment, pilgrimage and sacred sites have become a focal point for scholars (Okamoto 2015; Horie 2017; Carter 2018; Yamanaka 2020; Tillonen 2021). Some researchers have advocated the idea that pilgrimage and sacred places are typical examples of places that reflect the contested authority and legitimacy of institutionalized religions based on traditional doctrines and hierarchies (Okamoto 2015; Horie 2017; Carter 2018; Tillonen 2021). In this sense, pilgrimages and sacred spaces in contemporary society have dramatically declined to some extent, owing to the controversy of religious narratives in the phenomenon. As the tourism industry continues to expand around pilgrimages and sacred places, religious sites as a place for both religious and tourism activities have become an area of controversy, with narratives contested among different actors such as tourists and the tourism industry (Carter 2018; Tillonen 2021). Carter (2018) showed the conflicts surrounding the power spot phenomenon within the Association of Shinto Shrines (Jinja-Honchō), which is the largest religious association in contemporary Shinto established in 1946 (Carter 2018, pp. 161–63).

In contrast, other researchers have described a change in the form of religious legitimacy itself, owing to the transformation of the contemporary social environment, such as changes in individuals' lifestyles and worldviews (Reader 2012, 2013; Okamoto 2015, 2020; Yamanaka 2016, 2017, 2020; Tillonen 2021). Yamanaka (2020) showed that "contemporary religions can be analyzed by viewing them in the context of a globalized consumer society, the development of information technology, and market economy principles" (Yamanaka 2020, p. 5). Hence, the narratives around the power spot phenomenon and the social environment seem to be embodied in new understandings of religiosity, which can be called "spiritual legitimacy" in this study, according to the transformation of religious legitimacy embedded in Japanese society.

Moreover, the public discourses of the power spot phenomenon embodied the new landscape of sacred topography, and formed new forms of social movements and commitments to the sacred places, which have enhanced certain social ties and community relationship (Kato and Pregano 2017; Uchikawa 2017; Pregano and Kato 2020). While previous studies placed a strong focus on the individualization of religious commitments and the decline of religious communities (Horie 2009, 2017), public images and discourses of power spots and the commitment of visitors have become a crucial opportunity for the shrine and its local community to trigger the revitalization of community activities.

In the case of Haruna Shrine in Takasaki City, west part of Gunma Prefecture in Japan, the spread of power spot discourse has dramatically changed its social environment. While

the shrine and local community had seriously suffered from the rapid decline of pilgrims and residents in the area (Todokoro 2007; Nishino and Todokoro 2012), the spread of the power spot narratives have revitalized social practices as well as religious activities. As Haruna Shrine has frequently mentioned in social networking service and digital media, as well as travel magazines and guidebooks as "one of the leading power spots in the Kanto area" (Visit Gunma 2021), and the number of visitors to the shrine has risen since 2011, local community members have been actively encouraged to rebrand their community images, in corporation with the Haruna shrine, local administrations and tourism industry.

This study, therefore, examines the emergence of a new form of spiritual legitimacy utilizing a case study of the power spot phenomenon and the Haruna Shrine in Japan, how the spiritual narratives related to the power spot phenomenon are consumed among stakeholders, to define a new form of spiritual legitimacy in the field. The paper discusses the characteristics of power spot phenomenon, constructed social environments and emerging social contexts in the Haruna Shrine.

## 2. Materials and Methods

The methodology of the paper is based on an empirical case study utilizing a qualitative approach to content analysis (Neuendorf 2002; Krippendorff 2004), which highlights the historical development of the Haruna Shrine from the perspective of the power spot phenomenon, and highlighted historical transformation of the shrine and its community in Haruna Area. The data is based on the relevant documents published by the Haruna Shrine, local administrations, the travel media, scholars, and the field research conducted from December 2020 to January 2021. The author conducted field research to collect social images and symbols related to the spiritual narratives and to explain how these narratives were consumed and formed the contemporary landscape of Haruna Shrine and Haruna Shrine Town.

## 3. Power Spot Phenomenon and Spirituality Movement in Japan

The development of the power spot phenomenon in Japan is frequently described as a part of the so-called "spirituality movement" in Japan, which coincides with the decline of traditional institutional religions in contemporary Japanese society, and the rise of individualized way of piety and religiosity (Shimazono 1999; Horie 2009, 2017; Okamoto 2015, 2020; Kato and Pregano 2017; Yamanaka 2016, 2017, 2020; Yamaguchi 2017), as many Japanese describe themselves as "spiritual but not religious" or "spiritual but not affiliated" (Mercandante 2014; Yamanaka 2020, p. 11). The spirituality movement has gained popularity worldwide. Researchers dealing with Japanese religious studies have shown that the spirituality movement was transmitted to Japan following the development of the so-called New Age movement in the West in the 1970s (Shimazono 1999; Horie 2009, 2017; Yamanaka 2020). Consequently, the spirituality movement and non-institutional religions have gained popularity among Japanese people rather than institutional religions (Okamoto 2015; Horie 2017).

Researchers have summarized the characteristics of the spirituality phenomenon in Japan as providing individuals self-enlightenment and the fulfilment of their spiritual needs (Horie 2009, 2017; Kato and Pregano 2017; Yamaguchi 2017). Horie (2009) summarized the characteristic of spirituality in Japan as a way for people to satisfy their need to seek their spirituality with no permanent commitment to a particular religious group. Kato and Pregano (2017) further noted that "individuals have obtained freedom of choice in defining their own spiritual life; spirituality is a matter of personal choice, and the individual is free to practice different spiritual traditions as they see fit, no longer under dogmatic restrictions from external religious institutions" (Kato and Pregano 2017, p. 244).

Other studies have emphasized that the popularization of public legitimacy based on non-institutional religions in the society has coincided with the individualization of piety in terms of spirituality. Researchers posited that contemporary society has developed the popularization of spiritual consumption and commodification, which fits in contem-

porary consumer society, instead of the decline of traditional institutional religions in the society (Taira 2009; Reader 2012, 2013; Okamoto 2015, 2020; Yamanaka 2020). Taira (2009) stated that spirituality is firmly connected with capitalism and consumer society, and "commodification is that in spirituality old traditions are privatized or individualized by leaving questions of community and social justice off the agenda" (Taira 2009, p. 232). As Olsen (2013) posited, the development of religious consumption forms a certain religiosity that fits contemporary consumer society by taking into account marketing and consumer behaviors based on individual motivations, demands, and preferences. The development of religious commodification has, therefore, promoted a new form of legitimacy in which Reader (2013) indicated the "democratization" of religious legitimacy, through the development of consumption of spiritual narratives (Reader 2013, p. 93; Yamanaka 2020; Yasuda 2020).

Hence, the spirituality movement can be described as the decline of religious legitimacy based on the hierarchical structure of traditional institutional religions and authority, and it is also recognized as a new form of religiosity based on consensus-building through consumerism among stakeholders, including non-traditional communities in Shinto like commercial and tourism enterprises, media industry and local governments and administration, as well as tourists (Pregano and Kato 2020; Yamanaka 2020; Yasuda 2020).

In this situation, spiritual legitimacy is experienced through consensus-building among stakeholders, which can be described as the process of authentication. In the process of authentication, Cohen and Cohen (2012) conceptualised two modes of the authentication process—"cool authentication" and "hot authentication". The cool authentication refers to "a single, explicit, often formal or even official, performative (speech) act, by which the authenticity of an object, site, event, custom, role or person is declared to be original, genuine or real, rather than a copy, fake or spurious" (Cohen and Cohen 2012, p. 1299), based on certification and accreditation by authorities such as religious leaders, institutions, and governments. Conversely, hot authentication is described as "an imminent, reiterative, informal performative process of creating, preserving and reinforcing an object's, site's or event's authenticity" (Cohen and Cohen 2012, p. 1301). The process of hot authentication is "emotionally loaded, based on belief, rather than proof, and is therefore largely immune to external criticism" (Cohen and Cohen 2012, p. 1301), and involves "a high degree of commitment and self-investment on the part of the participants" (Cohen and Cohen 2012, p. 1301). In this sense, the power spot phenomenon can be considered the process of hot authentication to enhance individual commitment to a certain spirituality movement.

Although hot authentication is based on individual preferences, attitudes, and motivations, the process has also enhanced social consensus for spiritual legitimacy through public negotiation and communication among stakeholders (Schilderman 2011; Reader 2013; Olsen 2013, 2019; Yamanaka 2020; Yasuda 2020). Thus, contemporary religion "stipulates its character as a commodity that has a certain value in exchange, and therefore may contribute to some shared or common good" (Schilderman 2011, p. 46) in the process of hot authentication as social interactions.

In the case of Japan, sacred places like Shinto shrines and Buddhist temples and related pilgrimages began to connect with tourism and the media to promote their sacred sites and showcase their spiritual prowess that enable consumers to achieve spiritual fulfilment and wellness according to market principles (Reader 2013; Kato and Pregano 2017; Carter 2018; Yamanaka 2020). Although most of the Shinto shrines and Buddhist temples in Japan do not officially describe their sacred places as power spots for spirituality purposes, they do not actively deny the alternative narratives related to the power spot phenomenon and spirituality and, consequently, attract a wider range of visitors (Okamoto 2015; Kato and Pregano 2017). Accordingly, the concept of religious commodification through the power spot phenomenon has been widely accepted in contemporary Japanese society. Thus, spiritual legitimacy in the power spot phenomenon is expressed through the process of hot authentication among stakeholders in a consensus-building governance system.

## 4. The Haruna Shrine and Syncretism in Japanese Society

The Haruna Shrine, located in the northwest part of Takasaki City, west part of Gunma Prefecture, is one of the most famous Shinto shrines in Japan (Figure 1). Mount Haruna was traditionally recognized as a sacred place owing to its majestic landscape and geography with exotic rock cliffs, fountains, waterfalls, and large avenues of cedar trees and other vegetation (Haruna Shrine 2021). Its geographical characteristics have contributed to its reputation as a sacred place of spiritual resonance (Kurihara 2009; Haruna Tourism Association 2021; Takasaki Tourism Association 2021), and the shrine's location on the slopes of Mount Haruna has made it a destination for mountain worshippers and for those seeking to fulfil their individual supplications and achieve spiritual satisfaction. However, in the twentieth century, the shrine was unfamiliar to secularized Japanese society, and only a small number of devotional followers made their pilgrimage to the shrine, with no more than three hundred thousand visitors per year in the 2000s (Nishino and Todokoro 2012). Nowadays, the visitors are increasing, with more than half million in 2019 (Takasaki City Municipal Office 2021).

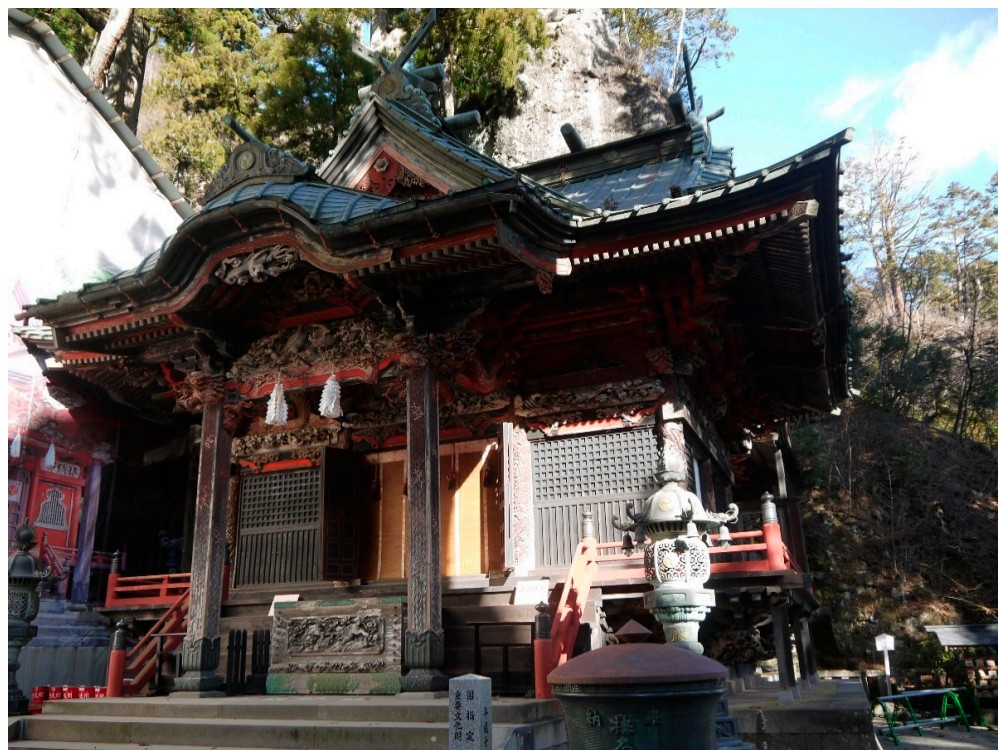

**Figure 1.** Honden (main building) and Misugata-iwa of Haruna Shrine. Photo by author on 19 December 2020.

Recently, despite its somewhat dilapidated environment until the 2000s, the Haruna Shrine has begun to attract people who are generally recognised as "non-religious" or "non-devotees" to a particular religion (Nishino and Todokoro 2012). Although the Haruna Shrine and its environment have been clearly recounted in Shinto doctrine, these visitors described the shrine as a power spot or as a spiritual or sacred place. Official publications of Haruna Shrine do not declare the shrine as a power spot or with any spiritual connotation; however, tourism media such as travel guidebooks and magazines describe the shrine as "one of the leading power spots in the Kanto area" (Visit Gunma 2021). Moreover, although the Japanese constitution clearly declares the separation of religion and state, local administrations and public institutions such as the Takasaki City Municipal Office and the Takasaki Tourism Association describe the shrine as a leading power spot or sacred place in which visitors can receive blessings and achieve spiritual fulfilment in their

promotional materials and local revitalization activities (Takasaki City Municipal Office 2021; Takasaki Tourism Association 2021).

People believe that the Haruna Shrine originated in the sixth century, and the Japanese Emperor Yōmei built a Shinto shrine in the location in AD 586. Other historical manuscripts, such as the book entitled *Engishiki* (Procedures of the Engi Era) published in AD 927, described the Haruna Shrine as one of the leading shrines in Japan (Haruna Shrine 2021). Nowadays, the shrine is dedicated to Homusubino Kami (the god of fire) and Haniyama Himeno Kami (the god of earth) (Haruna Shrine 2021). However, as the syncretism between Shinto and Buddhism was widely accepted in medieval Japan, the Haruna Shrine was considered a Buddhist temple and was called the Gandenji Temple, which enshrined Manghō Gongen, or Shōgun Jizō (victorious Ksitigarbha), and other *gongens* related to Mount Haruna (Kurihara 2009, p. 26; Imai 2012). In fact, as the Gandenji Temple, the shrine was managed by Buddhist monks as Buddhist temple, and Shinto activity rarely existed in the area (Kurihara 2009; Imai 2012). Moreover, Mount Haruna and the Haruna Shrine were recognised as the place for *shugen-dō* (ascetic practices based on syncretic mountain worship), and various *shugensha* (ascetic practitioners for *shugen-dō*) conducted their training at Mount Haruna.

The unusual historical setting of the Haruna Shrine was derived from the syncretic religious environment of medieval Japanese society, which is named *shinbutsu shūgō* (syncretism of kami and buddhas) in Japanese (Hardacre 2017, p. 530). Shinto—based on Japanese animism and polytheism—and Buddhism—based on Indian religious philosophy—were strongly syncretized in medieval Japanese society. The Japanese believed that Indian Buddhist deities were manifested in the form of an indigenous *kami* (god) in Japanese society based on the *honji suijaku* (Shinto/Buddhist syncretism) theory (Hardacre 2017, p. 148), in which *hotoke* (Buddhas deities) appeared in the form of indigenous *gongen* as their avatars in Japanese society. In this environment, the Japanese believed that Buddhism and Shinto were strongly connected in their daily lives, and they were extremely dependent on the *gongen*, which syncretized *gongens* fulfilled people's supplications in their lives (Hardacre 2017, p. 140). Sacred places such as Japanese Shinto shrines and Buddhist temples were based on the public belief of *gongen* in the medieval period, and other sacred places in the country were also syncretized.

In the Edo Period (from AD 1603 to AD 1868), people began to organize *kō* (a public group for pilgrimages) for pilgrimages to shrines and temples to demonstrate their particular devotion, and the development of these organizations further popularized pilgrimages (Hardacre 2017, p. 189). For example, Ise Jingū (Ise Grand Shrine) in the contemporary Mie Prefecture, the Kumano Sanzan Taisha (the sacred three shrines on Mount Kumano) and the Kōyasan Temples in the contemporary Wakayama prefecture, and other major shrines and temples are strongly connected with local community *kō* organizations to attract devotees to their shrines and temples (Hardacre 2017, p. 189).

As belief in the *gongen* was widely accepted in the medieval period, especially in the Edo Period, pilgrimage to the Haruna Shrine became a famous travel destination for the faithful as well as leisure for visitors. People in Edo City (now part of Tokyo Metropolis) and other areas organized Haruna Kō (public groups for Haruna pilgrimage) and annual pilgrimages to the shrine (Nagumo 1977; Imai 2012; Nishino and Todokoro 2012). With the development of Haruna Kō, the Haruna Shrine attracted pilgrims, and formed religious institutions and *shaya machi* (shrine town) to support their religious services (Nagumo 1977, pp. 130–31). *Oshi* (religious guides) organized religious tours and services for Haruna Kō and its members and developed a travel infrastructure for the pilgrims, such as *shukubō* (accommodations for pilgrims), dining venues, trails, and religious ritual spaces. Haruna Shrine established a large shrine town named Haruna Shaya Machi (Haruna Shrine Town), which became the base for pilgrims and other visitors (Nagumo 1977; Todokoro 2007; Nishino and Todokoro 2012). At its height in the nineteenth century, more than 200 *shukubō* and *oshi* attracted pilgrims (Nagumo 1977, pp. 132, 139–43), of which some 10 *shukubō*

remain as accommodation or restaurants and no *oshi* is active in 2020 per the author's field research (Figure 2).

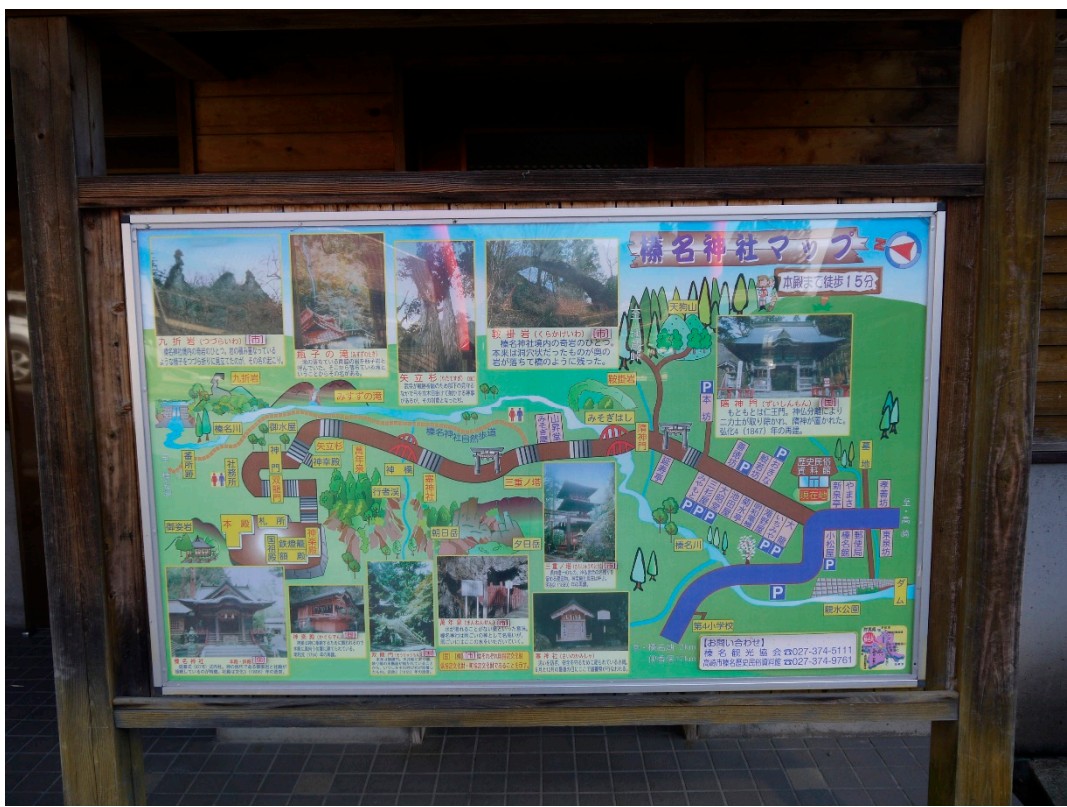

**Figure 2.** "Haruna Jinja Mappu (Haruna Shrine Map)," guide plate for visitors at Haruna Shrine Town by Haruna Tourism Association. Location and names of buildings in Haruna Shrine (left side) and accommodations and shops in Haruna Shrine Town (right side) are written in Japanese. Photo by author on 19 December 2020.

Despite being a renowned religious destination in the Edo period, pilgrimage to the Haruna Shrine and Haruna Shrine Town radically declined after the Meiji Restoration in 1868 AD (Nagumo 1977, p. 137; Imai 2012, pp. 58–84). The Meiji government heavily promoted imperial restoration based on Shinto, and consolidated Kokka Shinto (State Shinto), in which the government regulated Shinto and controlled its religious legitimacy through hierarchical religious institutions within the government (Hardacre 2017). The Meiji Government also promoted to remove other religions in Japanese society in order to enhance modern theocracy. In this situation, the new government released the Shinbutsu Hanzen Rei (The Order of Separation of Kami and Buddha) in April 1868. After the declaration of the separation of Shinto and Buddhism, Buddhism was widely abolished in Japanese society in the Haibutsu Kishaku (Abolishment of Buddhism) (Hardacre 2017). The Haruna Shrine was ordered to eliminate any Buddhist influences and to transfer its management to Shinto priests (Kurihara 2009, p. 27; Imai 2012, pp. 58–84). As a result, Shinto priests began to manage the shrine in 1872, and *gongen* and other Buddhist deities were removed from the space (Nagumo 1977, p. 137; Imai 2012, pp. 591–92).

Moreover, after the Meiji government ordered a ban on private figures to conduct Shinto religious services, the *oshi* and others related to the Haruna Shrine pilgrimages transferred their activities to restaurants and other leisure enterprises in the Haruna area (Nagumo 1977, p. 133). After the suppression of the Haruna pilgrimages, the Meiji government was highly involved in the control of religious legitimacy, such as the management and narratives of the Haruna Shrine (Nagumo 1977, p. 137; Imai 2012). The Haruna Shrine relied on the public institutions related to mass Haruna pilgrimages, and historical docu-

ments show that these changes completely destroyed its religious activities, landscapes and popularity, which were based on the medieval Japanese syncretism tradition.

## 5. The Power Spot Phenomenon and the Revitalization of Local Community in the Haruna Shrine

In the beginning of the 2000s, local residents of the Haruna Shrine area lamented its devastation, with obvious decline in visitation to the shrine by visitors and local residents (Todokoro 2007, p. 37; Nishino and Todokoro 2012), and recently the Haruna Shrine has experienced a resurgence of its popularity and legitimacy in modern Japanese society. The revival of the Haruna Shrine has primarily been driven by the media. Some Japanese geomancers introduced Haruna Shrine as "the place with spiritual power" in the mid-2000s (Voice Style 2006; Nihon Keizai Shinbun 2011), and women's lifestyle magazines, travel magazines and guidebooks began to promote the Haruna Shrine as "the leading power spot in Kanto area" (Asahi Shinbun 2010), or as a "power spot with sacred atmosphere" (Takasaki City Municipal Office 2012). Moreover, individual visitors began to share their spiritual journeys and experiences at the Haruna Shrine in digital spaces such as blogs and social networking services, posting new images and stimulating users' motivation for visiting the shrine, and articles in magazines and guidebooks promoted the Haruna Shrine as a place where supernatural forces originated from the religious activities and geography related to Mount Haruna and the Haruna Shrine (Kurihara 2009, p. 48; Takasaki City Municipal Office 2012; Haruna Tourism Association 2021). As power spot narratives spread around the Haruna Shrine, the number of visitors to the shrine almost doubled (Asahi Shinbun 2010; Jōmō Shinbun 2011; Nihon Keizai Shinbun 2011).

The narratives of spiritual visitors to the Haruna Shrine prefer to refer to and imitate previous experiences and narratives to ensure the spiritual legitimacy of their experiences and interpretations. Although Mount Haruna as a whole is recognised as a power spot destination, visitors discover their individual power spots, and find that rock cliffs, fountains, waterfalls, trees, bridges, and trails become their personal power spots. Although people may not share a specific image, symbol, or other knowledge of what a power spot is, there is a shared consensus among visitors regarding what symbolises these power spot narratives at Haruna Shrine. For example, visitors recognise Mount Haruna's famous rock cliffs, Misugata Iwa—behind the main building of the shrine, Nuboko Iwa—behind the Soryu-mon entrance gate, and other locations as the most powerful spiritual spots of the Haruna Shrine (Voice Style 2006; Kurihara 2009, pp. 41–44). Stakeholders share these symbols, developing legitimacy for the power spot phenomenon and validity for individual experiences and interpretations. These spiritual experiences were shared in digital spaces through SNS and other digital devices, which led to the process of hot authentication as Erik Cohen and Scott Cohen conceptualized (Cohen and Cohen 2012).

As the power spot phenomenon has developed, stakeholders began to construct and consume a narrative about the phenomenon and to participate in the phenomenon. For instance, travel companies and private travel groups organized packaged tours for tourists to experience the Haruna Shrine power spot and the shrine's spiritual atmosphere. Moreover, various social associations also organized excursions and tours to visit Haruna Shrine and Haruna Shrine Town to enhance their fortunes as well as their leisure purpose. The narratives promoted spiritual tours designed to foster the participants" good fortune and blessings, usually called *goriyaku* or *kaiun*, by experiencing the Haruna Shrine power spot (Voice Style 2006; Takasaki City Municipal Office 2010).

*Gūji* (Shinto shrine custodian) of the Haruna Shrine do not deny the power spot narratives—rather, they actively utilize these narratives to promote the shrine's association with Shinto perspective. For example, a Shinto priest at Haruna Shrine told an interviewer of the Asahi Shinbun newspaper that the "Shinto shrine itself owns the power of *kami*, and everywhere in the shrine reflects their powers" (Asahi Shinbun 2010). He also mentioned that he was surprised about the rapid increase of visitors to the shrine as the result of the power spot phenomenon (Nihon Keizai Shinbun 2011).

Although the revitalization of Haruna Shrine and Haruna Shrine Town were based on the development of the public discourses and social images of power spot phenomenon in the shrine (Figure 3), Takasaki City administration as well as the Haruna Shrine Town have contributed to the crucial role for the development (Nihon Keizai Shinbun 2011). As Haruna Town, where Haruna Shrine was situated, was merged with Takasaki City in 2006, the community development strategy and tourism strategy radically changed (Takasaki City Municipal Office 2010, p. 56; Nihon Keizai Shinbun 2011). After the annexation, the new Takasaki City adopted a new tourism strategy to revitalise Haruna Shrine and the Haruna Shrine Town, and created a new city image of Takasaki City (Takasaki City Municipal Office 2010, p. 21).

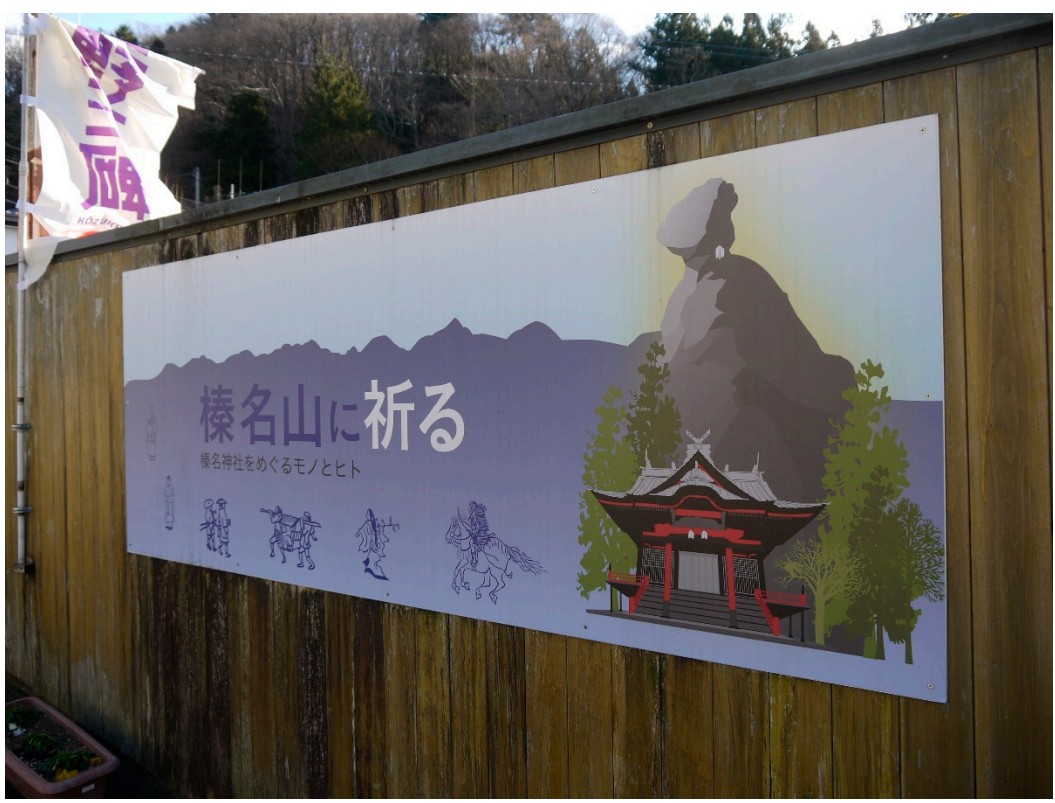

**Figure 3.** Banner "Harunasan ni Inoru: Haruna Jinja wo meguru Mono to Hito (Pray to Mt. Haruna: People and Objects at Haruna Shrine)" at Haruna Shrine Town. Photo by author on 19 December 2020.

As the power spot images and the symbols of the Haruna Shrine, like majestic landscape and geography, gained popularity, local administrations began to utilize public images and symbols based on spiritual narratives to achieve their own objectives for rebranding their public images of the city. Local administrations such as the Takasaki City Municipal Office and the Takasaki Tourism Association, and local communities such as the Haruna Shrine Town have rebranded their image to fit the power spot narratives, in cooperation with other sacred places in the city like Takasaki Byakue Dai-Kannon (Big Buddha Statue in Takasaki) and Shōrinzan Darumaji Temple, which is famous for the fortune Daruma named "Takasaki Daruma" or "Fortune Daruma" (CTBEC 2014). In fact, the Takasaki Tourism Association represents Haruna Shrine as "a popular power spot for visitors from around the country hoping to get a boost of good luck and fortunes" (Takasaki Tourism Association 2021). In this situation, Takasaki City introduced a new city image in its tourism strategy title "The Luck Town Takasaki (Engi no ii Machi Takasaki)," which states that the city historically enjoyed success and fortunes of social lives based on *engi*, *en* or *goen* (Japanese Buddhist term of *pratītyasamutpāda*, dependent origination or dependent arising) (Takasaki City Municipal Office 2012). In this context, sacred places in the city were

re-contextualised as living spiritual atmospheres of collective *engi* in contemporary social life (Figure 4). Hence, Haruna Shrine and Haruna Shrine Town, along with Big Buddha Statue in Takasaki, Fortune Daruma in Shōrinzan Darumaji and other sacred places in the city, has represented important place for *engi* and spiritual context in the city, which fulfil residents and visitors' business success and fortunes in their social life (Takasaki City Municipal Office 2012; CTBEC 2014, pp. 39–45). In this situation, Takasaki City and other administrative institutions have promoted tourism development projects and programs to improve tourism infrastructures, services and events in the Haruna Shrine and the Haruna Shrine Town to reflect new city image and narratives based on *engi* and spiritual narratives (Nihon Keizai Shinbun 2011; CTBEC 2014, pp. 43–44).

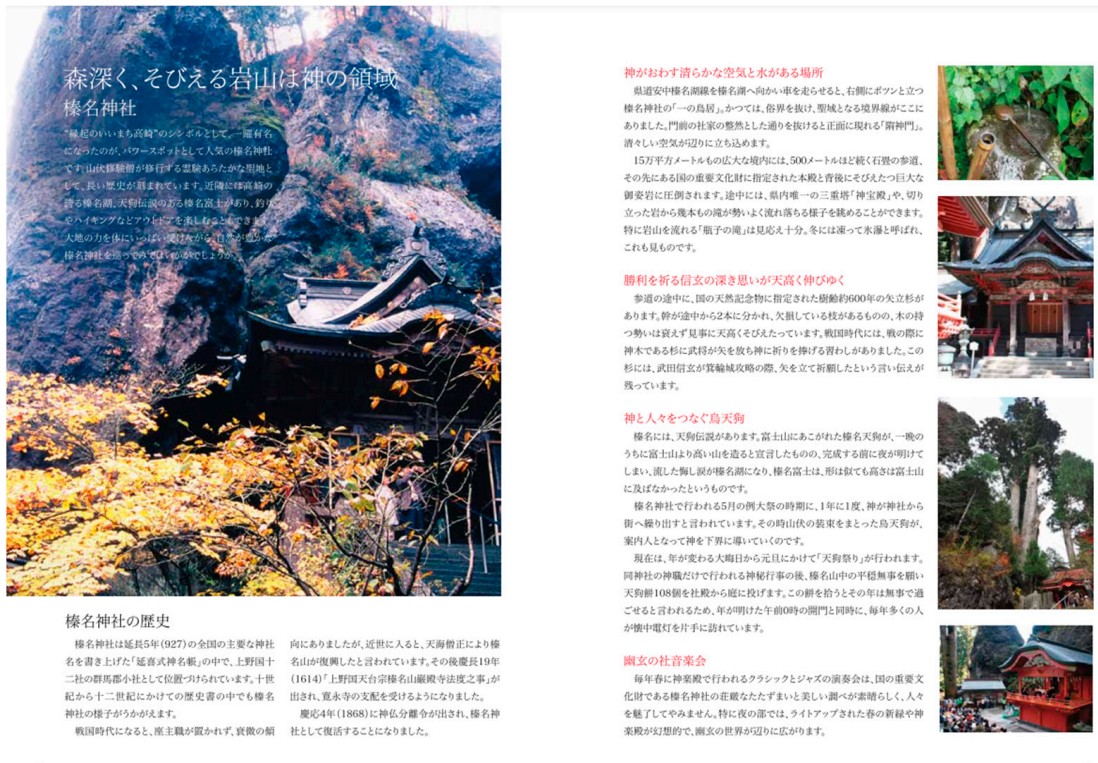

**Figure 4.** Explanation of History and Contemporary Significance of Haruna Shrine by Local Administration Guidebook. Source: (CTBEC 2014, pp. 44–45).

In this social setting, the Haruna Shrine and its local community have redefined their public image and social context based on consuming spiritual narratives. In fact, Haruna Shrine and Haruna Shrine Town have transformed its social context from religious, cultural and historical significance, which sustained the marks of local or discrete histories, in favor of aesthetic references to the desired or "correct" historical narrative or civilizational context of the local community (Todokoro 2007; Nishino and Todokoro 2012). Instead, they began to perform emerging social context as the continuity of *engi* and spiritual atmosphere, and commodified social context of spiritual narratives in the emerging public sphere related to the local community. In this context, Haruna Shrine and Haruna Shrine Town have actively connected their community activities with this social context for their revitalization. Although past contexts and practices have continued, this new context was situated in the center of its local landscape.

## 6. Making Spiritual Legitimacy in Power Spot Narratives

The example of Haruna Shrine and Haruna Shrine Town implies that individual experiences and narratives in the power spot phenomenon has promoted re-contextualization of

the place based on spiritual narratives in the sacred places, rather than de-contextualization of place and individualization of piety and narratives as held by previous literature (Horie 2017; Okamoto 2020). In this process, however, both host community and guest visitors struggle to confirm the authenticity and legitimacy of their spiritual narratives, through references to and imitation of perceived correct experiences and narratives, as Cohen and Cohen (2012) conceptualized in their discussion of the process of hot authentication. As a result, stakeholders have encouraged the formation of a certain social consensus, such as images and symbols, to legitimize their spiritual narrative and experience.

This hot authentication in power spot phenomenon has reformed and revitalized local communities through the development of public sphere for social context of spiritual narratives, which has developed further commitments and social interactions among stakeholders. As Kumi Kato and Nicolas Pregano indicated, the emerging spiritual narratives have heavily contributed to sustain and reform local community and members' social networks (Kato and Pregano 2017, p. 249; Pregano and Kato 2020). In fact, Haruna Shrine and Haruna Shrine Town have struggled to negotiate with emerging social environments and reformed their social context and images based on these spiritual narratives.

Hence, the spiritual narratives of the power spot phenomenon have contributed to developing social environments to embody social contexts based on spiritual narratives. As the discussion of religious commodification clarifies that spiritual legitimacy is embodied in the process of religious consumption, consumers are eager to construct a social context that embodies spiritual legitimacy for each experience and narrative by consuming similar images and symbols obtained by reference and imitation in the pilgrimage mobilities (Kitiarsa 2010; Olsen 2019). In fact, researchers have described how religious consumption promotes the involvement and interactions of new stakeholders—such as the tourism industry, governmental institutions and indigenous spiritual leaders, organizations and enterprises (Suzuki 2020), which leads to the "democratization" of religious legitimacy based on the social interactions (Reader 2013, p. 93). The development of hot authentication in the spirituality movement, therefore, has not only enhanced the individualization of piety and commitment but has also formed social consensus and public image through the development of a religious commodification environment or "spiritual market" (Yamanaka 2016, 2017, 2020).

Consequently, the development of the power spot phenomenon has created a new form of legitimacy based on consensus-building through the process of consumption, which can be named as "spiritual legitimacy." In this process, social environment and context for spiritual legitimacy are embodied in the commonly used images and symbols among the stakeholders, and the consumption of spiritual narratives embodies the spiritual legitimacy of the sacred places.

Although the development of the power spot phenomenon at the Haruna Shrine has superficially activated individualization of its spiritual legitimacy based on hot authentication, which encourages each visitor to interpret and utilize its legitimacy for their individual satisfaction, the phenomenon also constructs social environment and context for spiritual legitimacy among stakeholders by creating a new public sphere. In this sense, power spot narratives and practices have not only promoted individualization of piety but have also re-contextualized social images and meaning of sacred places and have made a certain community based on the consensus-building and legitimacy based on hot authentication.

## 7. Conclusions

This study examined the emergence of a new form of spiritual legitimacy utilizing a case study of the power spot phenomenon and the Haruna Shrine in Japan and explained how the spiritual narratives related to the power spot phenomenon are consumed among stakeholders to define a new form of spiritual legitimacy in the field.

The characteristics of power spot narratives emphasize the achievement of individual spiritual fulfilment and wellness in people's daily lives, such as healing, health, success and

good fortune. The constructed social environment for the power spot phenomenon does not directly assure the authenticity and legitimacy of individual experiences and narratives, owing to the absence of any firm legitimizing process and authority. However, this system reached an impasse with the transformation of the social environment in contemporary societies to form a certain social context through the development of social interactions among stakeholders. Consequently, people struggle to confirm the authenticity and legitimacy of their experiences and narratives in the process of hot authentication conceptualized by Cohen and Cohen. As a result, stakeholders have encouraged the formation of a certain social context, such as images and symbols, to legitimize these spiritual narratives and experiences. Although people's attitudes, motivations, and preferences are highly diversified owing to the diversity of their spiritual preference, attitude and motivation, the individual experience and the narrative of the power spot become synonymous because of the constructed social environment and context for the power spot phenomenon.

The emerging social context for the phenomenon, therefore, is constructed through the process of consumption of the spiritual narratives, which is reflectively socializing spiritual narratives through the individual consumption and social interactions with stakeholders. In the social environment of spiritual narratives, spiritual legitimacy strongly embodied a certain social context based on these spiritual narratives. In the case of the Haruna Shrine, specific activities and narratives gained social consensus, which symbolized the Haruna Shrine as a power spot, and stakeholders consumed these images for future consensus-building.

In conclusion, this new form of spiritual legitimacy is based on the social context, which embodies stakeholders' social consensus on spiritual narratives. Without an established religious authority in the field, people have struggled to construct a social context for spiritual legitimacy to ensure hot authentication of their individual narratives and experiences.

**Funding:** This research was funded by Grants-in-Aid for Scientific Research of JSPS (Japan Society for the Promotion of Science), grant number 18K18283 and 19H00564.

**Conflicts of Interest:** The author declares no conflict of interest.

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
