# Peer review of "Spiritual Legitimacy in Contemporary Japan: A Case Study of the Power Spot Phenomenon and the Haruna Shrine, Gunma"

_religions, doi:10.3390/rel12030177_

Round 1
Reviewer 1 Report
This is an interesting paper framing the "power spot" phenomenon within a larger phenomenon of religious consumerism and commodification. It also articulates the role and contribution of stakeholders in building consensus and authority, thus legitimizing its spiritual power through a process of religious commodification. The case of Haruna Jinja is an interesting one as it shows how a power spot narrative is constructed. This case shows the validity of the author's theoretical framework and thesis. The paper is well organized and sections offer a smooth transition from one another. The paper is overall well written.
However, there are three areas of improvements that I would like the author to address.
1) contextualization: the historical development of the power spot phenomenon is well summarized in ll.44 to 56. However, since this is the core of the paper, I feel it needs a more detailed contextualization before the discussion of its consistency with the pilgrimage and religious tourism industry that follows. As N. Horie (2017, The Making of power spot) suggests, for example, although the power spot phenomenon has been popularized since the 2010s, mostly by media, the term was well established among New Age practitioners since the 1980s (p. 193). Also, although power spots tend to be places historically connected to Shinto, sometimes they are not, such as in the case of Mt. Aso, Dewa Sanzan and others. These places are popular devotion pilgrimage destinations as much as they are tourist destinations, and it is precisely because of this ambivalence that they are so popular. The geography of Haruna Jinja, which sits on Haruna Mountain, triggered the revitalization project of local citizens who started constructing a power spot narrative (along with local revitalization activities) because of that. As mentioned by the author, the rigid constitutional state-religion division does not allow individuals to advertise or claim religious or spiritual goods, while they can advertise the beauty and value of their satoyama. Maybe more discussion on the ambivalence of these "power spots" would give more consistency to the thesis of consensus-building and religious commodification
2) Methodology:
- the paper draws upon Haruna Jinja, which is a good case. However, the author should make a stronger case on why choosing this shrine. Why is Haruna Jinja relevant as compared to other places? Whether the choice was based on the author's accessibility and proximity, or because Harina Jinja provides a better environment for investigation, the author should clarify the rationale.
- since ‘power spot’ is a popular phenomenon, and both legitimacy and consensus-building require the popular consensus, I would like to read qualitative data findings showing the stakeholders’ perspective: what makes Haruna a power spot and what do they think about the production and consumption of this commodity? I think that including qualitative data would greatly improve the originality of the paper, especially if discussed against the ‘non-religious’ context.
3) Sources and literature review. I would like to know more about the debate among Japanese scholars regarding the power spot phenomenon. The author relies more on English-written sources than Japanese sources: a more detailed review of the latter ones would improve the quality of the literature review.
- 197-198: ‘non-religious’. Please add references
- 288: in the mid-200s: reference needed
Author Response
Dear Reviewer,
Thank you so much for providing your comments and suggestions on the paper. I am also thankful for the time and energy you expended. My response to your comments and suggestions are as follow:
Point 1: Contextualization: the historical development of the power spot phenomenon is well summarized in ll.44 to 56. However, since this is the core of the paper, I feel it needs a more detailed contextualization before the discussion of its consistency with the pilgrimage and religious tourism industry that follows. As N. Horie (2017, The Making of power spot) suggests, for example, although the power spot phenomenon has been popularized since the 2010s, mostly by media, the term was well established among New Age practitioners since the 1980s (p. 193). Also, although power spots tend to be places historically connected to Shinto, sometimes they are not, such as in the case of Mt. Aso, Dewa Sanzan and others. These places are popular devotion pilgrimage destinations as much as they are tourist destinations, and it is precisely because of this ambivalence that they are so popular. The geography of Haruna Jinja, which sits on Haruna Mountain, triggered the revitalization project of local citizens who started constructing a power spot narrative (along with local revitalization activities) because of that. As mentioned by the author, the rigid constitutional state-religion division does not allow individuals to advertise or claim religious or spiritual goods, while they can advertise the beauty and value of their satoyama. Maybe more discussion on the ambivalence of these "power spots" would give more consistency to the thesis of consensus-building and religious commodification
Response 1: Thank you so much for your valuable comments and further references. I totally agree with your valuable suggestion, especially the suggestion that ‘triggered the revitalization project of local citizens who started constructing a power spot narrative (along with local revitalization activities) because of that.’ I have also added the references and discussion of Norichika Horie and other researchers on power spot phenomenon, and clarified the point as you mentioned.
In the discussion in (Horie 2017) as you mentioned, spirituality movement and power spot phenomenon could be seen from 1980s, but I believe that the characteristics of phenomenon has dramatically changed as Horie mentioned. Although Horie’s literatures and other research like (Okamoto 2015) have strong emphasis on the individualisation of piety in Japanese society, I believe the individualisation of piety also triggered the socialisation of these discourses through the consumption of these narratives among various stakeholders. I believe this is main point of my paper, so I revised my discussion in this line.
As you mentioned the ambivalence between popular devotion and tourism destination in the case of Dewa Sanzan, the ambiguous characteristics of social environment in Japanese society have also promoted social recognition of emerging new narrative, and it gained social significance among Japanese society through consuming these narratives among stakeholders.
For the last point, because of the landscape of Haruna shrine and its local community is differed from the shared social image of Satoyama in Japanese society, the discourse of Satoyama is not told in the case of Haruna. Rather, people emphasise on the unique landscape, frontier of human society, and continuity of management by the shrine and local community, in order to sustain its spiritual landscape and topography.
I thank again for your stimulus comments and suggestion for the revision.
Point 2: Methodology: the paper draws upon Haruna Jinja, which is a good case. However, the author should make a stronger case on why choosing this shrine. Why is Haruna Jinja relevant as compared to other places? Whether the choice was based on the author's accessibility and proximity, or because Harina Jinja provides a better environment for investigation, the author should clarify the rationale.
Response 2: I completely agree with your suggestion, and I have added the general view and importance of Haruna Shrine in section 1. Also, I added other information of contemporary situation of Haruna Shrine and its local community in section 4 and 5, so please check the detail in paper.
Point 3: Since ‘power spot’ is a popular phenomenon, and both legitimacy and consensus-building require the popular consensus, I would like to read qualitative data findings showing the stakeholders’ perspective: what makes Haruna a power spot and what do they think about the production and consumption of this commodity? I think that including qualitative data would greatly improve the originality of the paper, especially if discussed against the ‘non-religious’ context.
Response 3: Thank you so much for your comments. I have added further detail data in section to 5. So, please check the detail in section 5. Moreover, as mentioned response 1 and 2, I revised my discussion as I added another data in section 5.
Point 4: Sources and literature review. I would like to know more about the debate among Japanese scholars regarding the power spot phenomenon. The author relies more on English-written sources than Japanese sources: a more detailed review of the latter ones would improve the quality of the literature review.
Response 4: I added both Japanese and non-Japanese scholars’ discussions on the power spot phenomenon in section 1 and 3, so please check the detail. Especially, I added the literatures written by Horie Norichika, Nakanishi Yuji, Caleb Carter and Miya Tillonen and other scholars
Point 5: 197-198: ‘non-religious’. Please add references, 288: in the mid-200s: reference needed
Response 5: I added references. I please check the paper.
Again, thank you for giving me the opportunity to strengthen my paper with your valuable comments and queries. I have worked hard to incorporate your feedback and hope that these revisions persuade you to accept my submission.
Sincerely yours
Reviewer 2 Report
I recommend expanding the section about religious commodification and spiritual legitimacy at Haruna Shrine (the first section labeled #5). This seems to be the main payoff for the article's case study, and it is too brief. More detail about these processes will make the paper more significant.
Author Response
Dear Reviewer,
Thank you so much for providing your comments on the paper. I am also thankful for the time and energy you expended. My response to the reviewer’s comments are as follow:
Point 1: I recommend expanding the section about religious commodification and spiritual legitimacy at Haruna Shrine (the first section labeled #5). This seems to be the main payoff for the article's case study, and it is too brief. More detail about these processes will make the paper more significant.
Responce1: As for the section 5, I added detailed narratives and discussion as you mentioned, and revised other sections. Especially, I added public discourses how stakeholders have utilised the spiritual narratives and story telling practices in the case of Japan.
I would like you to check the revised version of the paper.
Sincerely yours
Reviewer 3 Report
There have been several studies of the "power-spot" phenomenon and this paper could (pending on substantial revisions) contribute to that literature, but the authors need to rethink the main argument and the way both secondary and primary data are presented.
Below, I offer some suggestions for improvement.
1) in the field of religious studies, the argument of "commodification of religion" has become obsolete because it reifies old essentialist ideas of "religion" as a "pure"/"authentic" phenomenon that has become "soiled" by the market. Recent studies such as those below have, however, shown like other human activities, "religion" cannot be thought outside of our consumer/market society, and that there has never been a time when both consumer needs and ideological needs did not inform religious practice. Did not pilgrims to Haruna shrine in the Edo period also buy talismans and engage in other consumer activities?
some useful references:
- Taira, T. (2009), ‘The Problem of Capitalism in the Scholarship on Contemporary Spirituality’, in T. Ahlbäck (ed.), Postmodern Spirituality, 230-244, Turku: Donner Institute.
- T. Martikainen and F. Gauthier (eds.), Religion in the Neoliberal Age: Political Economy and Modes of Governance. London: Routledge.
- F. Gauthier and T. Martikainen (eds.), Religion in Consumer Society: Brands, Consumers and Markets. London: Routledge
2) apart from Okamoto, Suga and Yamanaka, there have been other important contributions to the study of the power-spot phenomenon, which put into question the degree to which this phenomenon "is widely accepted" as the authors of this paper argue. These studies, especially Caleb Carter's paper below, provide really models for how to tackle the phenomenon.
useful references:
- Carter, C. 2018. ‘Power Spots and the Charged Landscape of Shinto’, Japanese Journal of Religious Studies 45(1): 145-173
- several papers by Horie Norichika (堀江 宗正)
3) Although Ian Reader argues that secularization in Japanese society has significantly progressed since the 1980s, I do not think that he believes spirituality has replaced religion in some way. Please see his paper below:
Ian Reader. 2012. Secularisation, R.I.P.? Nonsense! The ‘Rush Hour Away from the Gods’ and the Decline of Religion in Contemporary Japan. Journal of Religion in Japan 1(1):7-36.
4) There are several expressions regarding "Shinto" (Shintoism is not a term that is usually employed by scholars) and "syncretism" that reflect relatively old arguments in religious studies. It would be difficult to cover here problems with the conceptualizations offered by the authors of this paper, but the paper below is a good start point:
https://networks.h-net.org/node/20904/discussions/837862/review-jolyon-thomas-studies-shinto
5) the paper lacks information on methodology: except for the months of December 2020 and January 2021, the reader does not know what the fieldwork actually entailed: who and how many people were interviewed, what material were consulted and if there were limitations to this methodology. This is quite important information because it enhances the legitimacy of the argument.
6) Similarly, sources such as Yamanaka 2020 or Nagumo 1977 are often cited, but there is no reference to specific page numbers. I believe a more thorough citation practice is highly needed in this paper.
7) please provide sources for generalizing observations made in lines 35,44,72-75, 170-172, 214-215. These arguments seem to lack sources to back them up.
Author Response
For Reviewer
Thank you so much for providing your stimulus comments and suggestions on the paper. I am also thankful for the time and energy you expended. My response to your comments are as follow:
Point 1: in the field of religious studies, the argument of "commodification of religion" has become obsolete because it reifies old essentialist ideas of "religion" as a "pure"/"authentic" phenomenon that has become "soiled" by the market. Recent studies such as those below have, however, shown like other human activities, "religion" cannot be thought outside of our consumer/market society, and that there has never been a time when both consumer needs and ideological needs did not inform religious practice. Did not pilgrims to Haruna shrine in the Edo period also buy talismans and engage in other consumer activities?
some useful references:
Taira, T. (2009), ‘The Problem of Capitalism in the Scholarship on Contemporary Spirituality’, in T. Ahlbäck (ed.), Postmodern Spirituality, 230-244, Turku: Donner Institute.
Martikainen and F. Gauthier (eds.), Religion in the Neoliberal Age: Political Economy and Modes of Governance. London: Routledge.
Gauthier and T. Martikainen (eds.), Religion in Consumer Society: Brands, Consumers and Markets. London: Routledge
Response 1: I really thank your valuable suggestions for the further references and discussion.
First, my discussion in the paper is not to show the dichotomy between ‘pure’ and ‘soiled’ perspectives religions in the process of religious commodification, or to show the importance of ‘pure’ authentication of religions in the contemporary Japanese society. Rather, my main observation in this paper is to consider the spread of consumption of spiritual narratives triggered by the transformation of social environment in contemporary Japanese society such as the change of local community and the development of tourism. Thus, as you mentioned, my observation is also correspond with ‘both consumer needs and ideological needs inform religious practice’ in the case of contemporary Japanese society (and I also consider, it has continued throughout the history of any religion as well as Japanese religions).
In this sense, I have added Taira’s discussion, and changed my discussion and conclusion from the discussion of religious commodification, into the consumption of spiritual narratives in the case of Japanese local community. Thus, in the process of revision, I will omit most of the discussion in religious commodification (in section 1, 3 and 5). In this sense, I do not mention T. Martikainen and F. Gauthier’s two books in my paper. However, these two books and related literatures and discussions are really interested me and important references for my future research, so I will deepen my observation through these two books.
Point 2: apart from Okamoto, Suga and Yamanaka, there have been other important contributions to the study of the power-spot phenomenon, which put into question the degree to which this phenomenon "is widely accepted" as the authors of this paper argue. These studies, especially Caleb Carter's paper below, provide really models for how to tackle the phenomenon.
useful references:
Carter, C. 2018. ‘Power Spots and the Charged Landscape of Shinto’, Japanese Journal of Religious Studies 45(1): 145-173
several papers by Horie Norichika (堀江 宗正)
Response 2: I really thank your comments and suggestions for references. I have added the suggested references, and also added other ones to strengthen my discussion.
Also, as you mentioned in the discussion in (Horie 2017), spirituality movement and power spot phenomenon could be seen from 1980s, but I believe that the characteristics of phenomenon has dramatically changed as Horie mentioned. Although Horie’s literatures and other research like (Okamoto 2015) has strong emphasis on the individualisation of piety in Japanese society, I believe the individualisation of piety also triggered the ‘socialisation’ of these discourses through the consumption of these narratives among various stakeholders. I believe this is main point of my paper, so I revised my discussion in this line.
Point 3: Although Ian Reader argues that secularization in Japanese society has significantly progressed since the 1980s, I do not think that he believes spirituality has replaced religion in some way. Please see his paper below:
Ian Reader. 2012. Secularisation, R.I.P.? Nonsense! The ‘Rush Hour Away from the Gods’ and the Decline of Religion in Contemporary Japan. Journal of Religion in Japan 1(1):7-36.
Response 3: I also agree with your opinion that Ian Reader is not mentioned spirituality has replaced religion in some way. So, I will revise discussion in section 1 and 3 to avoid not to confuse the readers. I think his discussion is more focus on the consumption of religions and religiosity in contemporary Japanese society, which reflects the social mentality and characteristics of Japanese society.
Point 4: There are several expressions regarding "Shinto" (Shintoism is not a term that is usually employed by scholars) and "syncretism" that reflect relatively old arguments in religious studies. It would be difficult to cover here problems with the conceptualizations offered by the authors of this paper, but the paper below is a good start point:
https://networks.h-net.org/node/20904/discussions/837862/review-jolyon-thomas-studies-shinto
Response 4: I really thank your comment and suggestion, and I have checked and revised the term in my paper. Please check the detail.
As for the discussion on Shinto/Shintoism, I agree with your suggestion, but it is too huge topic to examine the detail in a single paper. As your recommended reference describe the ‘ambiguous’ characteristics and various social images of Shinto/Shintoism, how to describe Shinto/Shintoism reflects the researcher’s academic ‘gaze’ to the topic. Thus, it should be my future study.
As for the term of ‘syncretism,’ I also hesitate to use the term because of your comment, but I do not conceive alternative term of ‘syncretism’ to express shinbutsu shūgō in English. Thus, although I know the term has negative sense, I use syncretism to express the phenomenon.
Anyway, I will deepen my observation through your suggested website and related reference. I really thank your valuable comment and further reference.
Point 5: the paper lacks information on methodology: except for the months of December 2020 and January 2021, the reader does not know what the fieldwork actually entailed: who and how many people were interviewed, what material were consulted and if there were limitations to this methodology. This is quite important information because it enhances the legitimacy of the argument.
Response 5: I agree with comment on my fieldwork at Haruna Shrine, so I have added the detail of my field research in section 2, and added some other information in figures in section 3, 4, and 5.
My field research is more focus on the collection of documents and social images of Haruna shrine and Haruna Shrine Town, how spiritual narratives are consumed by other stakeholders, and formed a certain social context (images and representations) in the process of consumption (mentioned in response 1 and 2), rather than clarifying the inner experience, of each visitor through the interview of visitors and related figures.
Point 6: Similarly, sources such as Yamanaka 2020 or Nagumo 1977 are often cited, but there is no reference to specific page numbers. I believe a more thorough citation practice is highly needed in this paper.
Response 6: I thank your suggestion. I added pages on citation. Please check the revised paper.
Point 7: please provide sources for generalizing observations made in lines 35,44,72-75, 170-172, 214-215. These arguments seem to lack sources to back them up.
Response 7: I added references. I please check the paper.
Again, thank you for giving us the opportunity to strengthen my paper with your valuable comments and queries. I have worked hard to incorporate your feedback and hope that these revisions persuade you to accept my submission.
Sincerely yours